# Work-Related Fatalities Involving Children in New Zealand, 1999–2014

**DOI:** 10.3390/children8010004

**Published:** 2020-12-24

**Authors:** Rebbecca Lilley, Bronwen McNoe, Gabrielle Davie, Brandon de Graaf, Tim Driscoll

**Affiliations:** 1Injury Prevention Research Unit, Otago Medical School, University of Otago, 9054 Dunedin, New Zealand; bronwen.mcnoe@otago.ac.nz (B.M.); gabrielle.davie@otago.ac.nz (G.D.); brandon.degraaf@otago.ac.nz (B.d.G.); 2School of Public Health, Faculty of Medicine and Health, The University of Sydney, Sydney, NSW 2006, Australia; tim.driscoll@sydney.edu.au

**Keywords:** injury, work, children, agriculture, farm, transport, occupational injury

## Abstract

In high income countries, children under 15 years of age are exposed to workplace hazards when they visit or live on worksites or participate in formal or informal work. This study describes the causes and circumstances of unintentional child work-related fatal injuries (child WRFI) in New Zealand. Potential cases were identified from the Mortality Collection using International Classification of Disease external cause codes: these were matched to Coronial records and reviewed for work-relatedness. Data were abstracted on the socio-demographic, employment and injury-related circumstances. Of the 1335 unintentional injury deaths in children from 1999 through 2014, 206 (15%) were identified as dying from a work-related injury: 9 workers and 197 bystanders—the majority involving vehicle crashes or being stuck by moving objects in incidents occurring on farms or public roads. Those at highest risk were males, preschoolers, and those of Māori or European ethnicity. Work made a notable contribution to the burden of unintentional fatal injury in children with most deaths highly preventable, largely by adult intervention and legislation. To address the determinants of child WRFI greater attention on rural farm and transport settings would result in a significant reduction in the injury mortality rates of New Zealand children.

## 1. Introduction

Unintentional injury is a leading cause of premature mortality in New Zealand children aged 1–14 years, accounting for two in five deaths in this age group [1,2,3]. Child injury mortality is inequitably distributed in New Zealand [3]. There are clear disparities for New Zealand’s indigenous Māori children who experience a rate of fatal injury 3.5 times greater than that for non-Māori children [1]. New Zealand’s child injury mortality rate is amongst the poorest amongst comparative Organisation for Economic Co-operation and Development (OECD) nations, being more than twice the rate of Sweden, the United Kingdom, Netherlands and Italy [4]. A 2009 report card scoring the adoption and implementation of evidence-based child injury prevention policies found New Zealand implemented only half of these interventional policies, again comparing poorly to comparative European OECD nations [5].

Children, although not traditionally thought of as part of the formal workforce in high income countries, do participate in work under less formal arrangements such as casual work during school holidays, part-time work after school, and work in family businesses. Additionally, they may be exposed to workplace hazards when they visit or live on worksites, such as farms. The only previous New Zealand study of child work-related fatal injury (WRFI) examined incidents from 1985 through 1998 [6]. This study reported child WRFI commonly occurred when children were bystanders to another person’s work process or activity, with the agriculture sector and farms in particular the dominant setting for these injuries [6]. Studies on child WRFI in other countries are also limited, although available evidence from Australia and the United States show a similar pattern as New Zealand, with the agricultural sector being the most common industry involved in child WRFI [7,8,9]. 

Child fatal injuries are therefore rarely studied in a work-related context, despite work contributing significantly to the burden of unintentional injury for this age group and such child-related deaths being included in workplace safety legislative protections. Child WRFI data are difficult to obtain from official workplace injury notifications or injury compensation claims, as these databases typically only capture those aged 15 years and older. Furthermore, injuries sustained in a work setting are not readily identifiable in external cause codes as defined by the International Classification of Diseases (ICD) framework [10] and while some could be identified using corresponding ICD place of occurrence and activity codes higher levels of use of “other specified” or “unspecified” categories mean these are less readily utilized in research [11]. In contrast, in New Zealand, coronial records provide a complete and comprehensive source of child WRFI because all deaths that are ‘sudden and unexpected’ are referred to Coroners to determine the cause and circumstances of death. This study, utilizing Coronial records, provides the most up-to-date and comprehensive information available on child WRFI in New Zealand.

The research aims to address the current deficit in knowledge about child WRFI nationally and internationally, capturing all child fatalities where a work exposure directly or indirectly contributed to the causes and circumstances of the fatal injury incident. All children who were fatally injured on a worksite, in a public place or on a public road as a result of employment (paid, unpaid or in-kind for family business) or due to another person’s work were included. This research will provide directions for preventive actions by using coronial case file data to establish a complete and comprehensive cohort of all WRFI in children from 1999 to 2014 in New Zealand.

## 2. Materials and Methods

Unintentional work-related fatal injuries of children aged less than 15 years were examined as part of a larger study of work-related injury fatalities in New Zealand [12,13]. In brief, potential WRFI cases with a date of death registration on or between 1 January 1999 and 31 December 2014 were identified using New Zealand’s Mortality Collection, the most complete data source for all New Zealand deaths, including work-related injury fatalities. Injury deaths were identified in the Mortality Collection as those with an underlying cause of death coded to an external cause in the International Classification of Disease (ICD-10-AM) range V01 to X59, X85 to Y34, Y85 to 86, Y87.2, Y87.2 and Y89.9 [10]. Linkage to coronial records held by the National Coronial Information System was undertaken with all corresponding coronial records reviewed for work-relatedness. Coronial files were found for 98% of all 1335 relevant external cause deaths for children.

A broad definition of work-relatedness, compared with official data definitions, was used to capture all child fatalities to which a workplace exposure contributed. The work-relatedness of a fatal injury event was decided based on whether the decedent, at the time of the fatal incident, was: working for pay, profit or payment in kind; assisting with work in an unpaid capacity; was engaged in other work-related activities even when on a break or away from the workplace; or was a bystander (as defined below) to another person’s work activity. All fatal injury cases determined to be work-related were broadly classified as one of the following.

Worker deaths: the decedent was fatally injured in the course of work duties in a workplace (referred to as workplace WRFI), or on a public road (referred to as work-traffic WRFI).

Bystander deaths: the decedent was not working but died as a result of someone else’s work activity regardless of fault (referred to as bystander WRFI). These deaths could be further classified as bystander deaths occurring on a public road (work-traffic bystander), at a work place (workplace bystander) or to students of primary school age or older where the incident occurred during school time or while they were performing a task directly connected with their course (students).

Rural deaths: the decedent was fatally injured on a rural workplace (farm) where the circumstances did not satisfy the worker or bystander definitions above. This group includes farm deaths in children where it was difficult to ascertain the relative contribution of work and non-work exposures.

Socio-demographic characteristics including age, sex and ethnicity were obtained from the Mortality Collection. Prioritised ethnicity was determined by categorising individuals with multiple ethnicity responses in the order of Māori first, then Pacific, Asian and finally European/Other to provide a single response as per Ministry of Health ethnicity protocols [14]. Small area geographical meshblocks were coded from the physical address where the injury incident occurred. Small area-level deprivation was then derived using the 2013 New Zealand Deprivation Index (NZDep), with deciles categorised into quintiles, with ‘1–2’ representing those living in the least and ‘9–10’ the most deprived areas [15]. Mechanism of injury, location of injury incident, and agency and industry of incident were obtained from coronial records. Standard coding frameworks including the Type of Occurrence Classification System (TOOCS) and the Australian New Zealand Standard Industry Classification (ANZSIC) were used [16,17].

To describe the burden and patterns of child WRFI, frequencies and percentages were calculated. The risk of child WRFI was calculated by age group, sex, ethnicity and deprivation using incidence rates per 100,000 person years with 95% confidence intervals (95% CI). Population estimates for children aged less than 15 from the 2000, 2006 and 2013 Census were obtained from Statistics NZ with denominators for non-census years estimated by linear interpolation and extrapolation. Data were analysed using Stata V13.1 SE [18].

To illustrate the most common circumstances of child WRFI, a series of narrative “profiles” were created using a combination of quantitative data (analysis described above) and qualitative analyses. Qualitative analyses used a thematic analytical approach examining what the decedent was doing prior to the injury incident, what went wrong to cause the injury and the cause of death.

Ethical approval for this study was granted by the University of Otago Human Ethics Committee (Ref 15/065), the National Coronial Information System (Ref NZ007), and Health and Disability Ethics Committee (Ref OTA/99/02/008/AM05).

## 3. Results

### 3.1. Child Work-Related Fatal Injuries

Of the 1335 injury deaths of New Zealand children less than 15 years of age from 1999 through 2014, 206 (15%) were classified as being work-related. This equates to an average of 13 deaths per year, or 1.54 deaths per 100,000 children per year (Table 1). The overwhelming majority (91%) of these WRFI were bystanders to another person’s work activity, with a slightly higher proportion of these fatalities occurring in workplaces (*n* = 93) compared with the work-traffic setting (*n* = 84). Of the few (*n* = 9) workers identified, one was aged 5–9 years and the remainder aged 10–14 years. A small number of deaths were children in rural workplaces.

### 3.2. Multiple Fatality Incidents

Among the 206 children who were fatally injured, 160 (78%) were involved in single fatality incidents, and 46 (22%) were involved in multiple fatality incidents, giving a total of 193 fatal incidents over the 16-year study period. Of the 46 multiple fatality incidents, 25 (54%) incidents included more than one child aged 0–14, primarily involving two child fatalities; however, one incident involved three.

### 3.3. Socio-Demographic Characteristics

Regardless of work-related injury setting, it was more common for males (64%) to be fatally injured than females (36%), with the incidence of WRFI higher than that for females (1.9, 95% CI 1.5, 2.1 compared with 1.1, 95% CI 0.8, 1.3) (Table 2). The burden of child WRFI was split fairly evenly across the three age groups. Younger children aged 0–4 years were more commonly involved in workplace deaths (43%), while older children were slightly more likely to have sustained work injuries in the work-traffic setting (38% of 10–14 years). The incident rate was highest in 0–4 year-olds with 1.7 (95% CI 1.3, 2.1) deaths per 100,000 person years.

The greatest number of child WRFI was observed in children of European ethnicity, followed by those of Māori ethnicity. However, the rate of child WRFI was highest amongst Māori children (1.9 per 100,00 person years, 95% CI 1.1, 2.5) followed by those of European ethnicity (1.6, 95% CI 1.3, 1.9) Among deaths of Māori children, most occurred in work-traffic settings, while among those of European ethnicity the majority of deaths occurred in the workplace. There were no clear differences observed by level of deprivation other than the rate for the most deprived group of children (NZ Dep 9–10) being the highest.

### 3.4. Incident and Injury Characteristics

Over one third of all child work-related injury deaths occurred in the major industry grouping of agriculture, forestry and fishing (37%), almost exclusively involving agriculture (Table 3). Agriculture was the largest single industry group across all child age groups, with the largest contribution occurring in children under five years of age (40%). One in five child WRFI involved the ‘transport, postal and warehousing’ major sector (20%), predominantly in the transport sector. 

The most common incident locations of child WRFI were public roads (45%) and farms (35%). For those under five years of age, WRFI were more commonly sustained on farms (40%) while children aged 5–14 years were more likely to be fatally injured on public roads (51% in 5–9 years, 53% in 10–14 years). Fatalities occurring in industrial/construction settings were rare. Of the 71 children who sustained WRFI on farms, the majority normally resided on the farm where the incident occurred or in surrounding rural areas (*n* = 59, 83%). Children fatally injured while visiting farms were older (> 5 years) and most commonly operating farm vehicles, such as quad bikes. 

Overall, the most common activities at the time of the WRFI were being transported as a passenger in, or on, a vehicle (34%), followed by recreation/playing (26%). It was more common for children under five years of age to be fatally injured while engaged in recreation/playing (40%) activities than it was for older children. Almost 20% of older children (10–14 years) were fatally injured while driving vehicles.

The mechanism of injury most commonly involved in child WRFI was vehicle crashes (34%) and being hit by moving objects (32%) with these incidents occurring on- or off-road. Slightly more children under five years of age were fatally injured when hit by a moving object compared with vehicle crashes, while the opposite pattern was observed for older children.

### 3.5. Agency

The breakdown agent, which is the agent immediately triggering the fatal injury incident, was most commonly human behavior (39%) (Table 4). Over half of child WRFI in those aged 0–4 years involved human behavior: particularly the child’s own behavior, such as a being attracted to an agricultural pond but with, presumably, little cognitive awareness of the hazard presented by water; and preoccupied caregivers, such as working parents distracted by a work activity while supervising young children. The next most common breakdown agent was vehicles (31%), particularly involving cars and utility vehicles. Vehicles were the most common trigger for fatal incidents involving children aged 5–9 years (41%) and those aged 10–14 years (32%). Environmental factors were most common amongst children aged 10–14 years, frequently including sloping or rough ground surfaces which contributes to loss of control of a vehicle, such as a quad bike on a farm. 

### 3.6. Narratives of Common Circumstances

The majority of child WRFI incidents involved one of four narratives representing the most common and recurrent circumstances of fatal injury: 1) unsupervised young children on farms; 2) children operating a vehicle on farms; 3) children as vehicle passengers involved in a collision with a working vehicle; and 4) children hit as a pedestrian or cyclist by a working vehicle.

The first scenario includes distracted supervision of young children on farms. In this scenario, young children were often playing unsupervised in a secure area around a farm house or while accompanying working parents on the farm. The young child was able to leave the “supervised area”, a space often considered to be secure by the parent. The young child was attracted to a water feature, such as a stream or an unfenced agricultural pond, or towards animals, and was fatally injured. There was often a delay before the child is noticed missing, which points to the supervising parents having a high level of trust in the security of the space they place their child within and the difficulties in maintaining focus on child safety while engaged in work activities. 

Another common scenario was children operating adult-sized quad bikes on a farm for the common purpose of rounding up cows or other animals, or engaging in a farm-related recreational activity. In these situations, the child lost control of the quad bike while operating the quad bike. The cause of the loss of control was often due to an unexpected change in terrain or due to loss of traction on a sloping paddock. In all cases, the quad bike has flipped landing on top of the child, inflicting crush injuries. The child was generally unaccompanied by an adult and is found dead at the scene.

Working vehicles on public roads is also a common scenario involving child WRFI. The most common road-bystander scenarios involved children as passengers in a non-working vehicle where, in many cases, an adult driver made an error, such as a dangerous overtaking manoeuvre or crossing the dividing centre lines on a two-way road. The non-working vehicle invariably collided head on with an oncoming working vehicle, most commonly a heavy truck, resulting in substantial impact injuries in all occupants of the non-working vehicle. Another further common scenario involves children as pedestrians at intersections or crossing the road, or as a cyclist on public roads, being struck by a working vehicle. Often, the working vehicle in this scenario was a large truck where the child was located within the blind spot of the truck’s mirrors resulting in the driver being unaware of the presence of the child prior to the incident.

## 4. Discussion

Work-related fatal injury contributes substantially to the total burden of unintentional fatal injury in New Zealand; 15% of the total burden of fatal unintentional injury in children in New Zealand during the period 1999–2014 was attributable to a work exposure. Both the former Health and Safety in Employment Amendment Act 2002 and the new Health and Safety at Work Act 2015 require employers and the responsible Government Agencies to protect all people who come into contact with workplaces, including children [19,20]. As such, there is a clear legislative mandate to prevent WRFI in children and investigate the circumstances of any that do occur. This study’s findings indicate the highest risks for child WRFI are in males, those under five years of age and those of Māori or European ethnicity. Child WRFI most commonly involved vehicle crashes or being hit by moving objects in incidents occurring on farms or public road. The main industries involved included the agriculture and transport sector.

This is a largely hidden burden as New Zealand’s official WRFI data excludes bystander deaths, thereby excluding most WRFI deaths in children. This study’s novel data therefore lead to the identification of important missed opportunities to reduce the broader public health impact of work-related fatalities in New Zealand. Updated accurate data on the current patterns of work-related fatalities inform where workplace interventions to prevent fatalities to children are needed and allows for the monitoring in trends in child WRFI over time. Combined with previously published child WRFI data, this study has demonstrated a declining trend in child WRFI between periods. This study identified 1.5 (95% CI 1.3, 1.7) work-related fatal injuries per 100,000 children for the period 1999 to 2014, a considerable decrease compared to the rate of 2.1 (95% CI 1.8, 2.4) in the study that covered from 1985–1998 [6]. However, despite the decline, similar patterns of child WRFI to those described earlier remain: bystander deaths dominate and the agricultural industry is the most common industry in which child WRFI occurs [8]. While there was an overall decline in rates of WRFI in adult workers in New Zealand over a similar period, these trends have been variable [21]. For example, adult workers in the primary production sector of agriculture, forestry and fisheries experienced an increase in fatal injury risk over this time [21].

The majority of children who sustained WRFI were bystanders, consistent with patterns of work involvement observed in children in Australia [7,9] and, as would be expected, given the low proportion of children engaged in formal or informal work, especially at a young age. The preponderance of bystander deaths in children reflects inadequate control over the hazards generated by work activities largely under the direct control of an adult worker, thus providing insight into the failures in the control of hazards in these work situations. This study found it was rare for children less than 15 years to be fatally injured as a worker. The circumstances of work involvement change with increasing age. As children get older and their physical and cognitive abilities develop, there is increasing movement into the formal workforce which changes the exposure of the child to workplace hazards, and the pattern of fatalities becomes more like that of adults [21].

The age-related patterns of WRFI observed in this study are consistent with those observed elsewhere. The exposure of children to work hazards, even at a very young age, are partially explained by the norms and attitudes of parents. For example, children residing on New Zealand farms often accompany working parents, or are engaged in farm work, where there are distinctive patterns of exposure to farm tasks by age [22]. Families residing on farms can perceive that children raised on farms are more aware of hazards and are more capable of handling risky agricultural tasks even at a young age [23]. North American Guidelines informing adults of agricultural tasks appropriate to the different child physical and cognitive developmental stages and corresponding abilities of children have been shown to be effective at reducing the burden of injury in children on farms [24].

Males are commonly over-represented in child WRFI, particularly on farms [9]. Gendered patterns of exposure to work hazards occur on New Zealand farms, with male children more involved in farm tasks involving machinery and vehicles resulting in increased risk of WRFI for males [22]. Māori children are over-represented in unintentional fatal injury, accounting for half of all child unintentional injury deaths in New Zealand [25]. This study’s findings suggest injury prevention efforts to reduce child WRFI for Māori need to focus on reducing work-traffic crashes. Whilst child WRFI rates varied by level of deprivation, with the highest rate of child WRFI observed for the most deprived children, there were few other clear patterns observed. This is in contrast to the common trend of a gradient of risk of fatal injury with increasing levels of deprivation in child unintentional injury mortality overall in New Zealand [26]. Few other studies have considered this as a potential determinant of child WRFI.

Rural environments are an important determinant of child WRFI. The agricultural sector is consistently identified as being one of the highest risk groups for fatal injury in adult workers in New Zealand [27,28,29]. Others have previously identified children living on farms or rurally as carrying higher injury risks compared to other non-farming or urban counterparts [7,9,30,31]. In agriculture, unlike other industries, children provide informal labor for family farm operations at an early age and farms typically have dual purposes, being both a place of work and of residence. This poses higher risk of work-related injury for children residing on, or visiting farms, because they are exposed to common and high-risk hazards, such as farm machinery and moving vehicles [22], to which other children are not exposed.

Very young children are particularly vulnerable to WRFI on farms where high levels of active supervision are needed to keep them safe in a high hazard work environment. While a 2007/08 survey of NZ farms found it was uncommon for young children to accompany working adults on farms [22], when it does occur, it poses a particularly high risk of WRFI for young children. A lack of childcare options for rural farming families, and rural attitudes of including children in informal family farm work from a young age, have also been identified as a barrier to protecting farm children from the hazards of work [32]. Many farming parents in New Zealand feel that, in order to prevent injuries on farms, a child’s access to active farming areas, as well to farm machinery or vehicles, should be restricted, or at a minimum properly supervised by an adult [22]. Other opportunities for prevention include the use of the most suitable vehicle to transport children on farms and the use of child restraints, such as seat belts and car seats in work vehicles to ensure the safety of children as occupants in vehicles used on- or off-road [33].

Other work contexts were also identified as being of concern. This study identified that work exposures make a substantive contribution to 28% of the total burden of transport-related fatalities involving children aged under 15 in New Zealand. Using public roads is a necessary part of everyday life for children; therefore, it is unsurprising that motor vehicle traffic crashes (MVTC) are the overall leading cause of death in New Zealand children aged less than 15, accounting for over a quarter of all child deaths for the period 2006–2010 [34]. While our finding highlights the potential for reducing child fatalities on public roads through the influence of workplace safety policy, few of these incidents were triggered by a work vehicle, which limits the possibilities for prevention through work safety actions. One area of direct influence of work, however, that needs to improve is heavy vehicle driver awareness of cyclists and pedestrians. To help with these sensory technologies, alerting drivers of the presence of cyclists or pedestrians in blind spots should be adopted on working vehicles. The implementation of other evidence-based traffic safety interventions, such as proper child car restraints, use of seatbelts, alcohol controls, and improved road infrastructure, such as physically divided roads on routes carrying high volumes of heavy working vehicles, provide other opportunities to reduce the burden of child WRFI [35]. However, it is really only the last of these that is primarily a work-related prevention initiative. Fatalities occurring in industrial or construction settings that are more common amongst working adults [21] were rare in children.

It is important that children’s needs for preventive actions in the workplace are assessed alongside those of the working adult population because the patterns of child WRFI differ from those of adults. Simply applying injury interventions intended for adults may not adequately protect children. Children are considered vulnerable to WRFI due to their lack of power in the work organizational hierarchy, and their lack of physical and cognitive development, leading to children and adults underestimating risk [32]. Effective or promising strategies for preventing child injury broadly includes (listed from most to least effective) legislation, modification of the environment or a product, the use of safety devices, educational home visits, community based interventions and education and skills development [33,34].

New Zealand performs poorly in the prevention of child and adolescent fatal injury, with rates of child injury mortality ranking the worst out of out of 24 OECD countries [4]. A wide-ranging national strategy for child safety more broadly is currently lacking, with previous national strategic efforts, most notably the New Zealand Injury Prevention Strategy (operational from 2003 to 2013), falling victim to changes in political focus [36]. The Well Child/Tamariki Ora programme, which takes an integrated child health and development approach to improve child health and well-being, includes child safety. However, it predominantly targets home and transport-based injury risks [37]. Children are a notable omission from New Zealand’s Health and Safety at Work Strategy despite the strategy’s vision of “work [that] is healthy and safe for everyone in New Zealand” [38]. This situation serves to illustrate the lack of implementation of important and effective policy and legislative actions to improve safety for children [5]. Our study highlights the need for child safety strategies to address the substantial role of work exposures in addressing the burden of fatal injury for children in New Zealand. National strategies for farm safety have been developed in Australia; for example, farm focused interventions and education programmes have been implemented to address child safety issues on farms [39].

Overall, these findings identify where improvements in prevention efforts are needed to address child WRFI, serving to highlight the importance of managing work-related risks for children, particularly in the Agricultural sector. Most child WRFI are highly preventable, largely by adult intervention and enforcement of current workplace health and safety legislation. Furthermore, the recurrent common narrative scenarios point to the highly repetitive circumstances surrounding these incidents, implying that it is common to fail to learn from previous fatal incidents involving children. Interventions to address WRFI in the youngest children (0–4 years) should focus on improved adult supervision of children during play on farms, while interventions for children aged 5–14 years should focus on reducing vehicle crashes on public roads, alongside farm settings. To identify and address the hidden burden of child WRFI official data capturing work-related injuries should be expanded to capture fatal and non-fatal injuries sustained by children. Regular surveillance of the burden and patterns of child WRFI will directly inform interventions to change work practices harmful to children and allow for these incidents to be included in health and safety enforcement practice.

The availability of detail-rich Coronial records allowed for the comprehensive and accurate determination of work-relatedness and the collection of new information on the causes and circumstances of child WRFI unavailable from other data sources. The mandated requirement for all sudden and unexpected deaths to be notified to a Coroner for investigation means that virtually all child deaths due to external causes receive an inquest to determine the cause and circumstance of the fatal injury providing a comprehensive population basis for informing directions for intervention. The inclusion of common narratives capturing the recurring circumstances of injury is novel for studies describing the burden of child fatalities. This study is limited to 2014 as the most recent year available due to the lengthy time it takes for a Coronial inquest to be “closed” and become available for research purposes, limiting the currency of the data. It is a strength that this study expands the range of childhood work-related deaths to consider all industry groups with many previous studies focusing only on fatalities occurring on farms. This study is limited to fatal injury in children and the mortality rates for children fatally injured while working used the total child population due to the lack of working population denominators for children. The inclusion of traffic fatalities where the working vehicle was not “at fault” limits the generalisability to other countries with “at-fault” systems restricted to cases where there was liability on the part of a working vehicle. However, inclusion of cases regardless of fault in this study is consistent with New Zealand’s no-fault universal accident compensation system.

## 5. Conclusions

This study found that work makes a notable contribution to the total burden of fatal injury in children. Most child WRFI are highly preventable, largely by adult intervention and enforcement of workplace health and safety legislation. Greater attention on managing work-related risks for children, particularly in rural farm and transport settings, would result in a significant reduction in the injury mortality rates of New Zealand children.

## Figures and Tables

**Table 1 children-08-00004-t001:** Work circumstance of work-related fatal injuries to children (0–14 years), New Zealand, 1999–2014.

Work Circumstances	Frequency *n* (%)	Rate (95% CI) per 100,000 Person Years
Working	9 (4.4)	0.06 (0.03–0.12)
Bystander	187 (90.7)	1.39 (1.21–1.61)
Rural workplace	10 (4.9)	0.07 (0.04–0.13)
Total	206 (100.0)	1.54 (1.34–1.77)

Abbreviations: Rate—incidence rate per 100,000 person years, 95% CI—95% Confidence Interval.

**Table 2 children-08-00004-t002:** Socio-demographic characteristics by location of injury, work-related fatalities to children (0–14 years), New Zealand, 1999–2014.

Characteristics	Work-Traffic *n* = 96 *n* (%)	Workplace *n* = 110 *n* (%)	Total *n* = 206 *n* (%)	Rate (95% CI) per 100,000 Person Years
Sex				
Male	57 (59.4)	75 (68.2)	132 (64.1)	1.93 (1.54–2.18)
Female	39 (40.6)	35 (31.8)	74 (35.9)	1.14 (0.83–1.33)
Age group				
0–4 years	28 (29.2)	47 (42.7)	75 (36.4)	1.71 (1.35–2.14)
5–9 years	32 (33.3)	31 (28.2)	63 (30.6)	1.42 (1.09–1.82)
10–14 years	35 (37.5)	33 (29.1)	68 (33.0)	1.49 (1.16–1.89)
Ethnicity				
European	44 (45.8)	69 (62.7)	113 (54.9)	1.60 (1.32–1.92)
Māori	35 (36.5)	30 (27.3)	65 (31.6)	1.96 (1.15–2.50)
Asian	7 (7.3)	7 (6.4)	14 (6.8)	1.13 (0.61–1.90)
Pacific	9 (9.4)	4 (3.6)	13 (6.3)	1.05 (0.55–1.79)
Other	1 (1.0)	0 (0.0)	1 (0.5)	0.17 (0.04–0.90)
NZDep Index				
1–2 (least deprived)	14 (14.5)	13 (11.8)	27 (13.1)	1.11 (0.72–1.16)
3–4	20 (20.8)	19 (17.2)	39 (18.9)	1.66 (1.18–2.26)
5–6	10 (10.4)	16 (24.5)	26 (12.6)	1.03 (0.67–1.50)
7–8	13 (13.5)	27 (24.5)	40 (19.4)	1.43 (1.02–1.94)
9–10 (most deprived)	38 (39.5)	26 (23.6)	64 (31.0)	1.97 (1.52–2.52)

Abbreviations: NZDep Index—New Zealand Deprivation Index, Rate—incidence rate per 100,000 person years, 95% CI—95% Confidence Interval

**Table 3 children-08-00004-t003:** Industry, location, activity and mechanism of work-related fatal injuries to children (0–14 years), New Zealand, 1999–2014.

Characteristics	0–4 Years *n* = 75 *n* (%)	5–9 Years *n* = 63 *n* (%)	10–14 Years *n* = 68 *n* (%)	Total *n* = 206 *n* (%)
Industry of incident				
Agriculture, forestry & fishing	30 (40.0)	23 (36.5)	23 (33.8)	76 (36.9)
Transport, postal & warehousing	16 (21.6)	14 (22.2)	13 (19.1)	43 (20.9)
Arts & recreation	12 (16.0)	5 (7.9)	3 (4.4)	20 (9.7)
Education & training	0 (0.0)	3 (4.8)	10 (14.7)	13 (6.3)
Construction	3 (4.0)	5 (7.9)	3 (4.4)	11 (5.3)
Other	14 (18.7)	13 (20.6)	16 (23.5)	43 (20.9)
Location				
Public road	24 (32.0)	32 (50.8)	36 (52.9)	92 (44.6)
Farm	30 (40.0)	20 (31.8)	21 (30.9)	71 (34.5)
Home	8 (10.7)	0 (0.0)	0 (0.0)	8 (3.9)
School	2 (2.7)	3 (4.8)	3 (4.4)	8 (3.9)
Recreation/sport	7 (9.3)	1 (1.6)	2 (2.9)	10 (4.9)
Industrial/construction	2 (2.7)	3 (4.8)	0 (0.0)	5 (2.4)
Unspecified/other	2 (2.7)	4 (6.3)	6 (8.9)	12 (5.8)
Urban-rural indicator				
Urban	30 (40.0)	31 (49.2)	32 (47.1)	93 (45.2)
Rural	45 (60.0)	32 (50.7)	36 (52.9)	113 (54.8)
Activity				
Passenger, in/on vehicle	26 (34.7)	26 (41.3)	19 (27.9)	71 (34.5)
Recreation/playing	30 (40.0)	14 (22.2)	11 (16.2)	55 (26.7)
Driving vehicle	1 (1.3)	7 (11.1)	13 (19.1)	21 (10.2)
Riding (horse, bicycle)	0 (0.0)	6 (9.5)	10 (14.7)	16 (7.8)
Helping out	2 (2.7)	2 (3.2)	3 (4.4)	7 (3.4)
Other	16 (21.3)	8 (12.7)	12 (17.6)	36 (17.5)
Mechanism				
Vehicle crash	21 (28.0)	26 (41.3)	24 (35.3)	71 (34.5)
Hit by moving object	27 (36.0)	20 (31.8)	20 (29.4)	67 (32.5)
Electricity, drowning	17 (22.7)	7 (11.1)	8 (11.8)	32 (15.5)
Vehicle rollover	4 (5.3)	6 (9.5)	9 (13.2)	19 (9.2)
Other and multiple	6 (8.0)	4 (6.4)	7 (9.3)	17 (8.3)

**Table 4 children-08-00004-t004:** Agency triggering the work-related incident (breakdown agent), among children (0–14 years), New Zealand, 1999–2014.

Characteristics	0–4 Years *n* = 75 *n* (%)	5–9 Years *n* = 63 *n* (%)	10–14 Years *n* = 68 *n* (%)	Total *n* (%)
Vehicle	15 (20.0)	26 (41.2)	22 (32.3)	63 (30.5)
Animal/Human behavior	41 (54.6)	24 (38.0)	16 (23.5)	81 (39.3)
Environment	10 (13.3)	7 (11.1)	18 (26.4)	35 (16.9)
Manufactured material	1 (1.3)	4 (6.3)	3 (4.4)	8 (3.8)
Machinery/tools	6 (8.0)	2 (3.1)	6 (8.8)	14 (6.7)
Other/unspecified	2 (2.6)	0	3 (4.4)	5 (2.4)

## Data Availability

Data were obtained from a secondary provider and are not publicly available.

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
