# Peer review of "Work-Related Fatalities Involving Children in New Zealand, 1999–2014"

_children, 2020, doi:10.3390/children8010004_

Round 1

Reviewer 1 Report

Thank you for your submission, my concerns are as follows:

  1. Kindly check all the references word by word and line by line- there are plenty of errors in format, spellings, font size and italics, etc- align with journal guidelines.
  2. Tables need clearer description and demarcation of variable categories. 
  3. Tables should be stand alone, with footnotes on what means what and description of abbreviations.
  4. The data is very old, please include data till 2019.
  5. The analyses is mostly descriptive- please conduct more sophisticated data analyses.
  6. There are no major implications for practice and research or indications of novelty of the paper- kindly reflect on these aspects and how this paper can be used by various sectors.

Author Response

Reviewer 1

1. Kindly check all the references word by word and line by line- there are plenty of errors in format, spellings, font size and italics, etc- align with journal guidelines.

RESPONSE: The references have been checked against the journal guidelines as requested – the references are consistent with the journal guidelines given at https://www.mdpi.com/journal/children/instructions and https://www.mdpi.com/authors/references.

In reviewing the reference list we also found a couple of duplicated references (reference 4 was a duplicate of reference1 & reference 38 was a duplicate of reference 5).  The duplicate references have been removed and all references renumbered in the body of the manuscript and the reference list.

2. Tables need clearer description and demarcation of variable categories. 

RESPONSE: Demarcation lines have been added to Tables 1-3 consistent with the table style in recent “Children” publications.

Table titles have been made more clear (see lines 132, 153-154, 175-176, 221-222).

LINE 132 Table 1. Work circumstance of work-related fatal injuries to children (0-14 years), New Zealand,

LINE 153-154 Table 2. Socio-demographic characteristics by location of injury, work-related fatalities to children (0-14 years), New Zealand, 1999-2014.

LINE 175-176 Table 3. Industry, location, activity, and mechanism of work-related fatal injuries to children (0-14 years), New Zealand, 1999-2014.

LINE 221-222 Table 4. Agency triggering the work-related incident (breakdown agent), among children (0-14 years), New Zealand, 1999-2014.

3. Tables should be stand alone, with footnotes on what means what and description of abbreviations.

RESPONSE: Footnotes describing abbreviations used for 95% Confidence Intervals, incidence rates, and the New Zealand Deprivation Index have been added where relevant to Tables 1 and 2.

4. The data is very old, please include data till 2019.

RESPONSE: These data represent the most current, available data at the time of conducting this study.  Due to substantial delays in the Coronial process, available Mortality Collection data is at least 4 years old when it is first made available.  Time is then required to undertake comprehensively review and analyse the relevant Coronial records. This is clearly a limitation of using these data and this has been added to the discussion of limitations (Lines 414-416).

LINES 414-416 “This study is limited to 2014 as the most recent year available due to the lengthy time it takes for a Coronial inquest to be “closed” and become available for research purposes, limiting the currency of the data.”

5. The analyses is mostly descriptive- please conduct more sophisticated data analyses.

RESPONSE: A priori this is a descriptive study.  Descriptive analyses are adequate to answer the research question posed and represent progression of knowledge in this under-researched area of child injury.

6. There are no major implications for practice and research or indications of novelty of the paper- kindly reflect on these aspects and how this paper can be used by various sectors.

RESPONSE: To address this comment, and a similar comment from Reviewer 2, we have added a new paragraph to the paper (Lines 393-405) which adds further consideration of the implications of this study’s findings to those already included elsewhere in the discussion (such as Lines 356-358).  The strengths of the study now include consideration of the novel aspects of including qualitative narratives.

LINES 393-405: “Overall these findings identify where improvements in prevention efforts are needed to address child WRFI, serving to highlight the importance of managing work-related risks for children, particularly in the Agricultural sector. Most child WRFI are highly preventable, largely by adult intervention and enforcement of current workplace health and safety legislation. Furthermore, the recurrent common narrative scenarios point to the highly repetitive circumstances surrounding these incidents, implying that it is common to fail to learn from previous incidents involving children. Interventions to address WRFI in the youngest children (0-4 years) should focus on improved adult supervision of children during play on farms, while interventions for children aged 5-10 years should focus on reducing vehicle crashes on public roads, alongside farm settings. To identify and address the hidden burden of child WRFI official data capturing work-related injuries should be expanded to capture fatal and non-fatal injuries sustained by children. Regular surveillance of the burden and patterns of child WRFI will directly inform interventions to change work practices harmful to children and allow for these incidents to be included in health and safety enforcement practice.”

LINES 412-413: “The inclusion of common narratives capturing the recurring circumstances of injury is novel for studies describing the burden of child fatalities.”

LINES 284-286: “This study’s novel data therefore lead to the identification of important missed opportunities to reduce the broader public health impact of work-related fatalities in New Zealand.”

Reviewer 2 Report

Dear Dr. Rebbecca Lilley,

Thank you for the opportunity to review this manuscript.

Manuscript ID children-1038972 entitled “Work-related fatalities involving children in New Zealand, 1999­–2014” has been reviewed.

The authors present a retrospective nationwide study of pediatric work-related fatalities in New Zealand during 1999­–2014. While the findings presented in this study seem to be of interest to the audience of this journal, it is my opinion that some major revisions, specified in the following technical comments to the author.

Major points:

  1. P4 L157­–P6 L180: There is difficulty in understanding the results in this section, because the sequence of the presentation of the results is different as shown in Table 3 and the main text. The authors should present the results according to the order displayed in Table 3.
  2. P6 L182–P 190 and Table 4: The authors indicate that there is an “age-related pattern” in Tables 2 and 3. If available, I strongly recommend presenting the agencies for each age group to reveal the “age-related pattern” more clearly.
  3. P6 L191–P P7 219: I could not understand the significance of the supporting data of your argument. The data and method are not convincing. Please elucidate any data that would serve as basis in the Results section and the approach method in the Method section. Moreover, the data presented would cause confusion to the readers since the results and discussion are merged in this part. Please show the result and discussion separately.
  4. The authors indicated that the main objective was to provide directions for preventive actions in the Introduction section. Therefore, I strongly recommend that you clarify the age-related patterns and risk factors in pediatric work-related fatalities and specify the appropriate injury prevention for each age group based on your results.
  5. P9 L316–339: This study is descriptive and your expression is redundant. Please discuss the injury prevention based on your results, as mentioned above.

Minor points:

  1. There are some listing mistakes in Table 2. Please revise those.
  2. There are different terms used in the main text and table, such as playing and recreation/playing. I recommend using unifying expressions, so that understanding the contents of your manuscript becomes easier.

Hopefully my comments will help to improve your manuscript.

Author Response

Reviewer 2.

The authors present a retrospective nationwide study of pediatric work-related fatalities in New Zealand during 1999­–2014. While the findings presented in this study seem to be of interest to the audience of this journal, it is my opinion that some major revisions, specified in the following technical comments to the author.

Major points:

1. P4 L157­–P6 L180: There is difficulty in understanding the results in this section, because the sequence of the presentation of the results is different as shown in Table 3 and the main text. The authors should present the results according to the order displayed in Table 3.

RESPONSE: The results as described in Lines 157-180 (now Lines 168-205) have been reordered to reflect the ordering of results in Table 3 as requested and now take the following order: 1) industry of incident; 2) location; 3) urban-rural indicator; 4) activity; 5) mechanism.

2. P6 L182–P 190 and Table 4: The authors indicate that there is an “age-related pattern” in Tables 2 and 3. If available, I strongly recommend presenting the agencies for each age group to reveal the “age-related pattern” more clearly.

RESPONSE: Table 4 has been reconfigured to now include data by age-group with the addition of three new columns, alongside a total column, consistent with Table 2 & 3.  To ensure confidentiality and adhere to our ethical approvals the data have now been collapsed into main categories (Vehicle, Animal/Human behaviour, Environment, Machinery, Other/Unspecified). The results (Lines 207-216) now present the age-related patterns of the breakdown agent presented in the revised Table 4.

LINES 207-216: “The breakdown agent, that is the agent immediately triggering the fatal injury incident, was most commonly human behavior (39%) (Table 4). Over half of child WRFI in those aged 0-4 years involved human behavior: particularly the child’s own behavior, such as a being attracted to an agricultural pond but with, presumably, little cognitive awareness of the hazard presented by water; and preoccupied caregivers, such as working parents distracted by a work activity while supervising young children. The next most common breakdown agent was vehicles (31%), particularly involving cars and utility vehicles. Vehicles were the most common trigger for fatal incidents involving children aged 5-9 years (41%) and those aged 10-14 years (32%).  Environmental factors were most common amongst children aged 10-14 years, and most commonly included sloping or rough ground surfaces contributing to loss of control of a vehicle, such as a quad bike on a farm.”

3. P6 L191–P P7 219: I could not understand the significance of the supporting data of your argument. The data and method are not convincing. Please elucidate any data that would serve as basis in the Results section and the approach method in the Method section. Moreover, the data presented would cause confusion to the readers since the results and discussion are merged in this part. Please show the result and discussion separately.

RESPONSE: Qualitative narratives were included to provide deeper understanding of the largely repetitive and recurrent circumstances of child WRFI.  We have clarified this in the methods (Lines 115-118)

LINES 115-118: “To illustrate the most common circumstances of child WRFI a series of narrative “profiles” were created using a combination of quantitative data (analysis described above) and qualitative analyses.  Qualitative analyses used a thematic analytical approach examining what the decedent was doing prior to the injury incident, what went wrong to cause the injury and the cause of death.”

We have also clarified this in the results outlining the narrative scenarios (Lines 220-221).

LINES 220-221“The majority of child WRFI incidents involved one of four narratives representing the most common and recurrent circumstances of fatal injury”

Further reflection on these common circumstances have also been added to the discussion (Lines 393-395).

LINES 396-398: “Furthermore, the recurrent common narrative scenarios point to the highly repetitive circumstances surrounding these incidents, implying that it is common to fail to learn from previous incidents involving children.”

4. The authors indicated that the main objective was to provide directions for preventive actions in the Introduction section. Therefore, I strongly recommend that you clarify the age-related patterns and risk factors in pediatric work-related fatalities and specify the appropriate injury prevention for each age group based on your results.

RESPONSE:  Table 4 and Lines 207-216 now clarify the age-related patterns of the breakdown agency as requested in Point 2 by this reviewer.  The discussion has been clarified to identify age-related injury prevention foci where supported by our findings.

LINES 399-401: “Interventions to address WRFI in the youngest children (0-4 years) should focus on improved adult supervision of children during play on farms, while interventions for children aged 5-10 years should focus on reducing vehicle crashes on public roads, alongside farm settings.”

5. P9 L316–339: This study is descriptive and your expression is redundant. Please discuss the injury prevention based on your results, as mentioned above.

RESPONSE: (Note lines 316-339 – now 369-392).  We have balanced this comment against feedback from our two other reviewers.  We feel these two paragraphs support our research findings by adding further discussion on the potential mechanisms and context by way of an explanation of our descriptive findings.  We agree that the injury prevention aspects of the discussion needed greater emphasis, as also identified by reviewer 1. While specific injury prevention implications have been included throughout the discussion we have added a paragraph considering the implications overall (Lines 390-403).

LINES 393-405: “Overall these findings identify where improvements in prevention efforts are needed to address child WRFI, serving to highlight the importance of managing work-related risks for children, particularly in the Agricultural sector. Most child WRFI are highly preventable, largely by adult intervention and enforcement of current workplace health and safety legislation. Furthermore, the recurrent common narrative scenarios point to the highly repetitive circumstances surrounding these incidents, implying that it is common to fail to learn from previous incidents involving children. Interventions to address WRFI in the youngest children (0-4 years) should focus on improved adult supervision of children during play on farms, while interventions for children aged 5-10 years should focus on reducing vehicle crashes on public roads, alongside farm settings. To identify and address the hidden burden of child WRFI official data capturing work-related injuries should be expanded to capture fatal and non-fatal injuries sustained by children. Regular surveillance of the burden and patterns of child WRFI will directly inform interventions to change work practices harmful to children and allow for these incidents to be included in health and safety enforcement practice.”

Minor points:

1. There are some listing mistakes in Table 2. Please revise those.

RESPONSE: A few typographical errors have been corrected in Table 2 and demarcation lines have been added to Table 2 consistent with the table style in recent “Children” publications.

2. There are different terms used in the main text and table, such as playing and recreation/playing. I recommend using unifying expressions, so that understanding the contents of your manuscript becomes easier.

RESPONSE: The terms have been reviewed to ensure they are consistently used between the text and tables.

LINES 197-198: “Overall, the most common activities at the time of the WRFI were being transported as a passenger in, or on, a vehicle (34.5%), followed by recreation/playing (26%).”

LINES 198-200: “It was more common for children under five years of age to be fatally injured while engaged in recreation/playing (40%) activities than it was for older children.”

Reviewer 3 Report

Thanks to the authors for their important work on a sector of the population that is quite overlooked and under-represented in the occupational safety and health research - children.  As work in the informal sector increases, these data will be even more important to help design strategies to protect the health and safety of children. The study is well-designed and methods are appropriate for the significant aims.  While the data is specific to New Zealand, and the authors note the country's unique no-fault transportation policies, the study has important implications for other countries where consideration of child WRFIs is needed.  Focus on the design of appropriate preventive measures will be important to realize the potential impact of this study. 

Author Response

Reviewer 3

Thanks to the authors for their important work on a sector of the population that is quite overlooked and under-represented in the occupational safety and health research - children.  As work in the informal sector increases, these data will be even more important to help design strategies to protect the health and safety of children. The study is well-designed and methods are appropriate for the significant aims.  While the data is specific to New Zealand, and the authors note the country's unique no-fault transportation policies, the study has important implications for other countries where consideration of child WRFIs is needed.  Focus on the design of appropriate preventive measures will be important to realize the potential impact of this study. 

RESPONSE: Thank you for your positive review. No changes have been made on the basis of this review.

Round 2

Reviewer 1 Report

thank you for all the revisions.

Reviewer 2 Report

Dear Dr. Rebbecca Lilley,

Thank you for the opportunity to review this manuscript.

Manuscript ID children-1038972 entitled “Work-related fatalities involving children in New Zealand, 1999­–2014” has been reviewed again.

This second version of this manuscript has been much improved and will be of interest to readers of this journal.